# HBV-Integration Studies in the Clinic: Role in the Natural History of Infection

**DOI:** 10.3390/v13030368

**Published:** 2021-02-26

**Authors:** Teresa Pollicino, Giuseppe Caminiti

**Affiliations:** 1Laboratory of Molecular Hepatology, Department of Human Pathology, University Hospital “G. Martino” of Messina, 98124 Messina, Italy; 2Laboratory of Molecular Hepatology, Department of Human Pathology, University of Messina, 98124 Messina, Italy; giuseppecaminiti8@gmail.com

**Keywords:** hepatitis B virus infection, chronic hepatitis B, HBV DNA integration, HBsAg, HBx, oxidative stress, double-stranded DNA breaks

## Abstract

Hepatitis B virus (HBV) infection is a major global health problem causing acute and chronic liver disease that can lead to liver cirrhosis and hepatocellular carcinoma (HCC). HBV covalently closed circular DNA (cccDNA) is essential for viral replication and the establishment of a persistent infection. Integrated HBV DNA represents another stable form of viral DNA regularly observed in the livers of infected patients. HBV DNA integration into the host genome occurs early after HBV infection. It is a common occurrence during the HBV life cycle, and it has been detected in all the phases of chronic infection. HBV DNA integration has long been considered to be the main contributor to liver tumorigenesis. The recent development of highly sensitive detection methods and research models has led to the clarification of some molecular and pathogenic aspects of HBV integration. Though HBV integration does not lead to replication-competent transcripts, it can act as a stable source of viral RNA and proteins, which may contribute in determining HBV-specific T-cell exhaustion and favoring virus persistence. The relationship between HBV DNA integration and the immune response in the liver microenvironment might be closely related to the development and progression of HBV-related diseases. While many new antiviral agents aimed at cccDNA elimination or silencing have been developed, integrated HBV DNA remains a difficult therapeutic challenge.

## 1. Introduction

Hepatitis B virus (HBV) infection is a major global health problem. Worldwide estimates indicate that more than 2 billion people have serological evidence of previous or current infections with HBV, and that 257 million of this total are chronically infected (defined by hepatitis B surface antigen (HBsAg) positivity) [1,2]. Despite the existence of an effective vaccine and treatments, HBV infection is the cause of nearly 1 million deaths from liver disease each year [3]. HBV infection leads to a wide spectrum of liver diseases including acute hepatitis (AH), fulminant liver failure, chronic hepatitis, cirrhosis, and hepatocellular carcinoma (HCC) [4]. Over 90% of people infected in adulthood spontaneously recover from AH, which resolves within a few months with the loss of HBsAg and seroconversion to hepatitis B surface antibody (anti-HBs). Individuals who are unable to clear HBsAg (the hallmark of chronicity) develop chronic infection. By contrast, if the infection is acquired at birth, the risk of developing lifelong chronic HBV infection is 90%, and it is 16% to 30% if it is acquired in childhood [5,6,7]. In contrast to AH, characterized by robust and multispecific antiviral immune responses, which contribute to viral clearance, chronic HBV infection is characterized by an adaptive immune response that rarely results in virus elimination [8]. This failure may be due, in part, to the tolerogenic properties of the liver microenvironment that are able to suppress the antiviral immune responses [9,10], and to an impaired and dysfunctional HBV-specific T-cell response due to virus production of extremely high levels of HBsAg and hepatitis B e antigen (HBeAg), particularly in the early phase of the infection [8,11,12,13,14]. Once chronic infection is established, the inflammatory response sustained by antiviral T lymphocytes and non-antigen specific inflammatory cells can significantly increase the risk of developing liver disease including chronic hepatitis, cirrhosis, and hepatocellular carcinoma. The natural history of chronic HBV infection includes distinct phases (Figure 1), which differ from each other in relation to the levels of viral replication, viral antigen expression, and inflammation in the liver [15]. Annually, only about 1% of chronically HBV-infected subjects spontaneously reach a complete immune control of the viral infection, with HBsAg loss [7]. However, the clinical resolution of chronic infection or of acute hepatitis does not imply the complete eradication of the virus, as small numbers of HBV-positive infected hepatocytes can persist in these subjects [16,17,18,19,20] and may be the source of viral reactivation in the case of immune suppression [16,20,21]. In people who recover from acute infection, HBV persistence has been demonstrated by both the detection of HBV DNA in the liver [22,23] and the endurance of vigorous and functionally efficient HBV-specific T-cell responses, decades after the resolution of the infection [8,17,24]. The immune response is likely sustained by a continuing stimulation from small amounts of HBV antigens produced from minimal residual virus [17,18,25,26]. This is typical of occult HBV infection (OBI), which is defined as the persistence of replication-competent HBV DNA in the liver (that is, the cccDNA minichromosome) and/or HBV DNA in the blood of HBsAg-negative people [20]. There are peculiar characteristics of the HBV replication cycle that make virus eradication from the infected hepatocytes unreachable. During its life cycle, HBV generates a nuclear cccDNA minichromosome as well as integrated DNA sequences, which represent transcription templates for viral proteins [27,28]. The high stability of the HBV cccDNA minichromosome is responsible for the lifelong persistence of the virus in the nuclei of infected cells, and HBV proteins from integrated viral sequences appear to contribute to the long-term persistence of virus infection. HBV cccDNA is epigenetically regulated, and it is the only transcriptional template for the viral pregenomic RNA (pgRNA) and for all the messenger RNAs (mRNAs) that encode the structural capsid protein (the core antigen (HBcAg)); its secretory variant HBeAg; the polymerase; the large, middle, and small surface proteins; and the X protein [29,30,31]. All the three surface proteins include the HBsAg “a” determinant, which elicits the most effective neutralizing-antibody response upon vaccination or infection [32]. The HBx protein, with an N-terminal negative regulatory domain and a C-terminal transactivation or coactivation domain, is a multifunctional protein able to modulate gene transcription, signaling pathway activity, genotoxic stress responses, protein degradation, cell-cycle control, cell proliferation, and apoptosis [33,34]. These activities may directly or indirectly influence viral replication and may also be relevant to HBV-associated pathogenesis, especially hepatocarcinogenesis [34,35].

PgRNA, in addition to functioning as the mRNA for HBcAg and polymerase, is reverse transcribed within nucleocapsids into relaxed circular DNA (rcDNA). The nucleocapsids are then enveloped by the surface proteins and secreted from cells as a new infectious progeny virus or returned into the nucleus to replenish the intranuclear cccDNA minichromosome pool [29,30]. HBV DNA integration is not implicated in the production of new virions [29]. However, as a stable intrahepatic source of HBx transcriptional transactivator and of immunomodulatory HBsAg, it might favor viral replication, immune tolerance, and viral persistence [12,36,37,38]. The double-stranded linear DNA (dslDNA) is likely the primary substrate for HBV integration (Figure 2). It is an aberrant byproduct of virus DNA synthesis due to the in situ priming of viral plus-strand DNA [30,39]. Analogously to HBV rcDNA, the dslDNA can be either secreted from hepatocytes as new virions or recycled into the nucleus. Once the dslDNA is delivered into the nucleus, either after the de novo secondary infection of hepatocytes or intracellular nucleocapsid recycling, it may integrate at the sites of double-stranded DNA breaks of the host cell genome by nonhomologous end joining (NHEJ) or microhomology-mediated end joining (MMEJ) [40,41,42] (Figure 2). The sequence of the dslDNA implies that the integrated HBV DNA can, at most, transcribe one intact preS-S HBV mRNA, which may lead to the synthesis of surface proteins, which include HBsAg [12,37,38,43]. Preliminary data obtained by targeted long-read sequencing (Pacific Biosciences Sequel II platform) and RNA-Seq have shown that, in liver tissues from chronic hepatitis B (CHB), HBV integrations are not restricted to partial viral genomic regions but may also involve the full viral genome, and that most transcribed integrations contain HBV promoter sequences driving S transcripts expression [44]. Of note, it has been demonstrated that the HBsAg derived from integrated HBV DNA can contribute to the assembly and release of hepatitis delta virus (HDV), a satellite virus of HBV that requires HBsAg for the production of new virions [38,45]. HBV integration appears to occur at random sites of the host genome and has been detected in a vast number of different genes. It has been demonstrated that HBV integrates in early stages of infection, and this can culminate in HCC development after years of productive or occult infection [28]. It is of note that over the decades, available data on HBV integration have been acquired by applying multiple different integration-detection methods. These methods have progressed from the little-sensitive Southern blot hybridization techniques to the PCR-based methods and, more recently, to high-throughput sequencing technologies, which allow a refined examination of the integration events and of their potential clinical implications. Each of the methods has its distinct sensitivity, biases, and drawbacks [46] (Table 1), and, regrettably, in most of the studies, the underlying limits of the detection method used have not been taken into account during the interpretation of results.

## 2. HBV Integration in Acute B Infections

To date, there have been very few studies evaluating HBV integration in acute B infections. This is essentially due to the ethical barriers in obtaining liver tissue samples from patients with acute hepatitis. Some of the available data are from studies performed in the 1980s, in which virus integration was analyzed by applying the little-sensitive Southern blot hybridization technique, which at that time was the only available technique for the study of HBV integration. It requires the use of restriction enzymes, and provides no information concerning host–virus junction sequences [46]. By applying this technical approach, three studies demonstrated that HBV integrated sequences can be detected even in patients with fulminant hepatitis (FH) [70,71,72]. FH is a rare, dramatic clinical syndrome characterized by a sudden and massive destruction of hepatocytes, which leads to multiorgan failure in people with no evidence of previous liver disease [73,74]. Due to the extremely rapid clinical course of FH and the difficulties in obtaining liver specimens, data from the available clinical studies are of particular relevance. One of them analyzed liver biopsy samples from three children who had developed fatal FH, and who died 2–6 months after the onset of the disease [72]. Overall, the other two studies analyzed liver samples obtained by needle biopsy or autopsy from 27 adult patients with FH [70,71] an average of 23 days after the onset of symptoms [70]. Liver samples from all the three children and seven out of 27 patients with FH showed the presence of HBV DNA integration in the cellular genome even in the absence of free viral DNA [70,71,72]. Moreover, in most of the cases, HBV DNA appeared to be integrated at multiple sites in the host genome [70,72]. Altogether, the results from these studies demonstrate that HBV DNA integration occurs in the very first weeks of the infection [70,71,72]. Furthermore, two of these studies reported that most of the FH patients harboring HBV DNA integration into the cell genome were serum HBV DNA negative, had lost HBsAg, and showed the absence of HBV replicative intermediates in the liver [70,72]. This, therefore, indicates that cells with integrated viral sequences may survive the rapid immune destruction of infected hepatocytes. By using Southern blotting, HBV integration has also been detected in the liver tissues from patients with acute benign hepatitis [70,71]. These earlier observations have subsequently been confirmed by PCR-based approaches, which have also shown the presence of clonally expanded integration sites in the acute phase of infection. Therefore, besides demonstrating the occurrence of HBV DNA integration at a very early infection stage, there is also some evidence indicating that a sequence of events favoring the clonal expansion of liver cells carrying viral integration may take place during the short period of viral replication in AH [75,76]. The evidence of HBV integration in FH and AH is consistent with observations in animal and in vitro models, where integration has been detected shortly after infection [39,41,77]. In ducklings infected with the avian hepadnavirus duck hepatitis B virus (DHBV), integrated DNA was found as early as 6 days post-infection, with an estimated integration frequency of one viral genome per 10^3^ to 10^4^ cells [39]. In woodchucks, the insertion of woodchuck hepatitis virus (WHV) DNA sequences into the host genome was detected significantly earlier, at 1–3 h post-infection, demonstrating that WHV DNA integrates into the hepatocyte genome immediately after natural virus invasion. It was revealed that even infection with very small quantities of WHV (10 or 100 virions) leads to virus DNA integration into the woodchuck genome. In addition, integrated WHV DNA was found to persist in liver tissue from recovered animals at essentially constant levels of one viral genome per 1 × 10^3^–3 × 10^3^ hepatocytes, suggesting that liver cells in the recovered liver were primarily derived from the infected hepatocyte population. Indeed, as reported by Summers et al., during the recovery from woodchuck transient infection, cell turnover and compensatory cell proliferation may lead to the reduction of cccDNA-positive hepatocytes and, eventually, to the selection and accumulation of uninfected cells carrying integrated DNA [77]. This scenario is consistent with the observations in human liver chimeric mice that underwent serial transplantation with primary human hepatocytes (PHHs) from HBV-infected humanized mice. In the transplanted animals, the number of engrafted PHHs increased by 2–3 log over 100 days. High PHH proliferation correlated with a massive decrease in viral markers including cccDNA, but not with integrated viral sequences, which instead persisted. Moreover, in lamivudine-treated transplanted animals, integrated HBV DNA was detected in almost all the PHHs [78]. Thus, available data indicate that whereas cccDNA molecules do not survive mitosis, viral integration persists and may be favored during hepatocyte proliferation. The observation, that in animal models, integrated DNA can survive virus infection clearance, is in accordance with previously reported data from patients with FH, where it has been shown that HBV integration is not related to the duration of viral replication and that integrated HBV DNA represents the only persisting marker of HBV infection. In addition to these in vivo observations, different studies using various cell models of HBV infection (including PHH, HepaRG-NTCP, HepG2-NTCP, and Huh7-NTCP systems) have demonstrated that viral integration occurs soon after infection [41,79]. In particular, it has been shown that HBV integrates as early as 30 min post-infection into the genome of HepG2-NTCP-C4 cells [79]. Integration was detected in all the cell types at a rate higher than 1 × 10^4^ cells and could be efficiently blocked by treatment with the HBV entry inhibitor Myrcludex B [41], but not with nucleos(t)ide analogs (NAs) [80], interferon alpha, or the capsid assembly inhibitor GLS4 [41]. This indicates that HBV DNA integration happening shortly after cell infection is likely independent of de novo HBV genome replication in this model. Analysis of the genomic location of the integration sites showed that viral integrations were randomly distributed over the entire human genome. However, there is evidence indicating that transposable elements frequently represent the initial sites of HBV integration [79]. These elements are widespread in the human genome. Some of these elements are able to mobilize and can naturally “jump about” the genome. The insertion of mobile elements into the DNA of germ cells can disrupt genes, leading to the sporadic development of diseases, and their insertion into somatic cell DNA may contribute to tumorigenesis [81]. HBV DNA integration into a transposable element would suggest the spreading of HBV DNA across the host genome from the very early phases of viral infection, with the potential transposition of the chimeric HBV-retrotransposon sequences across chromosomes and within genes. Mobile elements are preferred sites for homologous or nonhomologous recombination, and the presence of cellular conditions during HBV infection that increase the rate of genomic instability may possibly favor the occurrence of a significant number of viral integration events in these sites [82]. In this context, by applying a newly developed algorithm (SurVirus) [83] enabling the improved detection of viral integration, Rajaby, R., et al. [83] reported that HBV integrations are particularly enriched in long interspersed nuclear elements (LINEs) and satellite regions. Interestingly, HBV integration in a host LINE with the generation of an HBx–LINE1 fusion transcript was previously reported in 21/90 HBV-related HCC patient tumors (23%), and was significantly associated with poor patient outcomes [68]. These data were confirmed in a different study from China (17/40 cases, 42.5%) [84], though the HBx–LINE1 chimeric transcript was not detected in HBV-related HCCs from Europe [85]. Therefore, the high frequency of this potentially oncogenic transcript might be restricted to HBV genotype C infection, which is highly frequent in Chinese patients, but it needs to be validated in different series of HCC from other geographic areas. From a mechanistic point of view, the HBx–LINE1 transcript was shown to act as a long non-coding RNA that activates Wnt/β-catenin signaling, thereby promoting HCC [86]. Indeed, it appears that HBx–LINE1 expression leads to the migration and invasion of tumor cells through the induction of epithelial–mesenchymal transition and the nuclear localization of β-catenin. Moreover, there is evidence indicating that HBx–LINE1 may act by causing the sequestration of miR-122 [84]. Indeed, a negative correlation has been found between HBx–LINE1 levels and miR-122 amounts in HBV-related HCCs [84]. HBx–LINE1 expression, by reducing the levels of functional miR-122, may potentially lead to the transcriptional activation of hundreds of miR-122-targeted genes and the impairment of hepatic homeostasis in HBV-related HCCs [84].

It is likely that the HBV integration identified in acute infections represents “passenger” events and that after AH recovery, most viral integrants are lost from subsequent hepatocyte generations [40,77]. However, the early occurrence and persistence of integration events after infection recovery may provide hypothetical evidence that integration could be implicated in the modulation of HBV activities (i.e., the activation or silencing of viral transcription/replication activities) and, possibly, in lifelong virus persistence. Indeed, the role of viral integration in the HBV life cycle still remains an unexplored field, which should be investigated.

## 3. Natural History of Chronic HBV Infection

The complex interplay between viral replication and the host immune control is responsible for the variable course of chronic HBV infection. There are five distinct phases in its natural history according to the European Association for the Study of the Liver 2017 guidelines [15], which do not always occur sequentially. These phases have been identified on the basis of specific biochemical, serological, and virological characteristics, including HBeAg status, serum HBV DNA, and alanine aminotransferase (ALT) levels (Figure 1) [15].

The first phase is characterized by HBeAg positivity, high levels of HBV replication, normal ALT, and minimal or no necroinflammation activity. This phase was previously termed “immune tolerant” on the basis of the absence of clinical and histologic evidence of liver disease despite the presence of a high HBV viremia titer (of ~1 × 10^9–10^ virions per mL). In people infected at birth, this phase is more common and prolonged. The notion of immune tolerance has recently been disputed. This is essentially related to reported evidence showing that the HBV-specific T-cell response in this phase has no significant difference from that observed in the immune-active phase of the infection [87,88]. Moreover, though cirrhosis does not occur in the immune-tolerant phase of infection, early stages of fibrosis have often been reported. Furthermore, it has been shown that events implicated in the initiation and promotion of hepatocarcinogenesis, such as HBV integration and clonal hepatocyte expansion, may start in this early phase of chronic infection [59]. On the basis of this evidence, it has been decided to rename this phase “HBeAg-positive chronic infection” [15]. The second phase, named “HBeAg-positive chronic hepatitis”, is characterized by the presence of HBeAg, and high serum HBV DNA and ALT levels. In the course of this immune-active phase of chronic infection, adaptive immunity and the recruitment of non-antigen specific inflammatory cells lead up to an inflammatory response, which if sustained, may lead to cirrhosis development. In contrast to acute infections, this immune response rarely results in viral clearance [8,15]. This inability may be caused in part by the capacity of the liver to suppress intrahepatic immune responses [9,10]. Additionally, in the early phase of infection, HBV chronicity may also be promoted by HBeAg. Indeed, mouse model studies indicate that this secretory viral protein may suppress HBV-specific adaptive immune responses to HBV infections acquired at birth from HBeAg-positive but not HBeAg-negative mothers [8,89,90]. ALT flares and the exacerbation of liver disease are also typically observed during this phase and may mirror the immune clearance of infected hepatocytes. Dramatic ALT elevation associated with a strong decline of HBV viremia levels is often observed in the transition to the immune-control phase and followed by HBeAg loss. Most patients who clear HBeAg enter the immune-control phase (that is, the ”HBeAg-negative chronic infection” third phase), but some progress directly to the “HBeAg-negative chronic hepatitis” fourth phase [15]. The “HBeAg-negative chronic infection” third phase, previously termed the “inactive carrier” phase, is characterized by the absence of HBeAg and presence of anti-HBe, low serum HBV DNA levels, and normal ALT. These patients have an excellent prognosis and a very low risk of liver disease progression if they have not developed significant liver damage before entering this phase and if they maintain the immune-control state [15]. However, liver disease reactivates in about a third of patients in the immune-control phase, with an increase in both ALT and serum HBV DNA levels above 2 × 10^3^ IU/mL. The “HBeAg-negative chronic hepatitis” phase is characterized by HBeAg negativity and anti-HBe positivity, moderate/high serum HBV DNA levels, and elevated ALT. Exacerbations of chronic hepatitis can be observed during this phase [91,92]. The immune-control and HBeAg-negative hepatitis phases are associated with the selection and emergence of HBeAg-negative HBV genomic variants, which have mutations in the precore and/or basal core promoter regions of the HBV genome that abolish or downregulate HBeAg production [38,93]. During the HBeAg-negative phases, HBsAg has been shown to be preferentially produced by integrated HBV DNA sequences, although cccDNA is detectable but at a lower level than in the HBeAg-positive phases [37]. The rebound of viremia above 2 × 10^3^ IU/mL could be related both to the epigenetic modification of the cccDNA minichromosome and to an increased number of hepatocytes that are replicating HBV. Even in the presence of normal ALT, HBV viremia levels above 2 × 10^3^ IU/mL are related to an ineffective immune control of the virus and are often an indicator of active liver disease [94], in particular, in HBeAg-negative CHB cases characterized by fluctuating biochemical and virological profiles over time [95].

The final phase is the “HBsAg-negative phase” corresponding to OBI, which may occur either spontaneously or under antiviral treatment, and it is associated with improved prognostic outcomes including a reduced (but persistent) risk of cirrhosis and the development of HCC [15,20]. The loss of HBsAg has also been defined as a “functional cure” to underline the achievement of effective host immune control of virus activities and the transcriptionally silent state of the HBV cccDNA [27,96,97]. Studies performed in chronically infected patients reported that viral DNA integration is an early event in HBV infection that can occur in all the phases of chronic HBV infection, and both in vitro and animal model studies have confirmed these findings [28].

## 4. HBV Integration in Chronic B Infections

Chronic HBV infection exposes patients to significant risks of liver disease development including chronic hepatitis, cirrhosis, and HCC. There is evidence indicating that HBV DNA integration may play a role in the progression of liver disease. Liver damage due to HBV infection is characterized by continuous necroinflammation modulated by immune cells [13,14]. As a precocious and persisting event in the course of HBV infection, viral integration appears to be strictly linked to the immune-mediated ongoing inflammatory response, which during CHB, leads to the oxidative damage of liver cells’ DNA [98,99] (Figure 2). This damage can lead to increased levels of double-stranded DNA breaks, which represent the substrate for HBV integration. Indeed, several reports have shown that oxidative damage causes an increased genomic amount of hepadnavirus integration in growing cell lines [100,101]. Therefore, double-stranded breaks in cellular DNA, resulting from inflammation-induced DNA damage and regeneration, would be reflected in the number of HBV DNA integration events in the infected liver. Moreover, hepatocyte regeneration in response to the cytotoxic T lymphocyte (CTL)-mediated killing of HBV-infected liver cells may favor HBV integration via clonal hepatocyte expansion. Indeed, HBV-specific CTLs selectively kill HBV-replicating hepatocytes, leading to the clonal expansion of HBV DNA-integrated hepatocytes, as they may escape the HBV-specific CTL response [58].

It has been well documented that HBV DNA integration and clonal hepatocyte expansion can be detected in all the five phases of chronic HBV infection [20,37,55,59,60,102,103,104,105,106,107,108]. However, data demonstrating the occurrence of HBV integration in the early phase of chronic infection have only recently been provided. This is essentially due to the fact that the first phase of chronic infection (the HBeAg-positive chronic-infection phase) has been considered to be benign in nature. Patients were viewed as disease-free, defined as immune tolerant, and excluded from therapy based on international guidelines. However, some authors have recently challenged the concept of immune tolerance, showing the presence in immune-tolerant patients of both HBV-specific immune responses and histologic features of active liver disease [59]. In these patients, the presence of HBV DNA integration and the clonal expansion of a large number of hepatocytes bearing integrated HBV DNA have also been demonstrated. Furthermore, HBV integrants in the host genome were observed at a frequency whereby integration events could mutate any host gene in at least one hepatocyte in the liver. Interestingly, the use of integrated HBV DNA as a cell lineage marker to quantify clonal hepatocyte expansion in immune-tolerant patients, and in age-matched immune-active noncirrhotic HBeAg-positive and HBeAg-negative CHB patients, has led to demonstrating that during the progression of chronic HBV infection, hepatocytes harboring integrated HBV sequences undergo large-scale selective clonal expansion. Indeed, the presence of a large number of hepatocyte clones could be observed both in immune-tolerant and in HBeAg-positive immune-active patients, and some of them could show clones with a high cell count (~1000 cells). In HBeAg-negative immune-active patients, hepatocyte clone sizes appeared to be even bigger in number (>10,000 cells), similar to those observed in HCC patients [59]. The mathematical modeling of normal liver regeneration has demonstrated that hepatocyte clones of such a large number were unlikely to be generated by a random nonselective turnover of hepatocytes [58], and that they probably reflect the expansion of hepatocytes with a selective advantage [8,59,60]. Therefore, the process of selective growth or survival advantage appears to be active across all the three studied phases of chronic infection. Of particular note is the striking expansion of hepatocyte clones with integrated HBV DNA in patients with HBeAg-negative chronic hepatitis. This finding is corroborated by evidence, both in humans and chimpanzees, demonstrating that HBeAg-negative cases have the highest number of HBV-integration sites [37,46,48,108]. These data would entail an enrichment of peculiar cellular genomic sequences at the sites of HBV integration during the strong clonal expansion of hepatocytes associated with HBeAg seroconversion. However, a recent study has refuted this possibility. Indeed, Budzinska et al. [104], analyzing 717 unique virus–cell junctions detected by inverse nested PCR (invPCR) in nontumor liver tissues from HBeAg-positive and HBeAg-negative patients, showed that in HBeAg-negative patients, the sites of HBV DNA integration were distributed across the host chromosomes with no evident enrichment of specific structural or functional cellular genomic sequences at viral integration sites. In addition, it was suggested that the large number of viral integrations detected in HBeAg-negative patients was not the cause but rather the consequence of the extensive clonal expansion of hepatocytes (>10,000 cells), which may be exclusively linked to liver regeneration induced by the antiviral immune response during HBeAg seroconversion. Thus, the authors indicate that the majority of HBV DNA integrations in CHB patients would act as passenger mutations and do not represent initiators of carcinogenesis through insertional mutagenesis or the cis-regulation of cellular genes [104]. Although the shortcomings of the invPCR assay (that is, (a) the use of restriction enzymes to recover the virus–host DNA junction (potentially favoring the recovery of integration sites closer to the same restriction sites in the host genome) and (b) the exclusive detection of integrations that occur between nucleotides ~1650 and ~1850 of the HBV genome [46]) may have affected the data obtained by Budzinska et al. [104], they are very similar to results from several studies that have analyzed nontumor liver tissues adjacent to HCC by applying next generation sequencing (NGS) approaches. Indeed, most NGS studies have evidenced strikingly distinct patterns of HBV integration between tumor and adjacent nontumor tissue samples even from the same patients [42,61,62,67,69], supporting the hypothesis that HBV integration likely plays a cis-mediated oncogenic role mainly during tumor progression. However, it has recently been shown that a high number of HBV integrations is associated with viral replication in nontumor liver tissues and that the number of HBV integrations is an independent prognostic factor in HBV-related HCCs [42]. It should, however, be underlined that the HBV integration profile of nontumor liver tissues from HCC patients may, to some extent, diverge from that of liver tissue from CHB patients. It is of note that, despite the progress in sequencing technology, genomic datasets from large cohorts of CHB patients are not available. Moreover, comparative studies evaluating HBV-integration events in the different stages of chronic liver disease, including cirrhosis by NGS approaches, have, to date, not been performed. This is mainly due to the fact that with the emergence of the noninvasive assessment of liver fibrosis and efficacious anti-HBV NA therapies, tissue sampling by liver biopsy is now rarely carried out [109]. Therefore, the role of integrated HBV DNA in clonal hepatocyte expansion and events leading to hepatocarcinogenesis in CHB patients require further investigation, underlining the need for more studies based on human tissue samples in this setting. By contrast, whole-genome and transcriptomic sequencing data from surgically dissected HBV-related HCC and paired adjacent nontumor liver tissue are widely available [61,62,63,66,67,69,110,111]. A recent study [112]—assuming that adjacent nontumor liver tissue samples from HCC patients can serve as surrogate samples for investigating the genetic and transcriptional profiles of CHB patients—collected whole-genome and transcriptomic sequencing data from two large genomic studies that evaluated patients with HBV-related HCC [61,62] to estimate the frequency of viral-integration events and the fraction of HBsAg expression from integrated HBV DNA in nontumor liver samples. This study estimated that the average HBV-integration frequency was 0.844 per diploid genome in the nontumor tissue samples and that ~80% of the HB transcripts were derived from integrated HBV DNA [112]. This last observation is consistent with findings from other studies supporting a significant contribution of integrated HBV to serum HBsAg levels in CHB patients [37,108] and in chimpanzees [37]. Interestingly, transcriptomic analysis of chronically infected HBeAg-positive and HBeAg-negative chimpanzees revealed that the vast majority of intrahepatic HBV transcripts from HBeAg-negative animals ended before the HBV polyadenylation signal, supporting the idea that these 3′-deleted viral RNAs (encoding HBsAg) were transcribed from integrated HBV DNA. By contrast, very few 3′-deleted transcripts were found in the HBeAg-positive animals. A remarkable hepatic HBsAg staining suggestive of the clonal expansion of hepatocytes in HBeAg-negative chimpanzees but not in the HBeAg-positive animals was also observed [37]. These data could be explained by significant HBsAg expression from integrated HBV in the absence of actively transcribed cccDNA in the same hepatocytes. In the chimpanzee animal model, it was also found that total liver HBV DNA was present at a lower amount in HBeAg-negative animals compared to HBeAg-positive animals, and that the level of liver HBV DNA was almost unchanged by therapy with NAs in HBeAg-negative chimpanzees [37]. These data indicate that most of all the HBV DNA in the livers of these HBeAg-negative animals was not representative of episomal viral DNA but rather of HBV DNA integrated into the host genome, thus strengthening the evidence that integrated HBV DNA is the main source of HBsAg in the HBeAg-negative phase of chronic hepatitis.

As highlighted previously, the HBeAg-negative phase of chronic HBV infection may be associated with different virologic/clinical outcomes including HBeAg-negative chronic infection (serum HBV DNA < 2 × 10^3^ IU/mL and normal ALT) and HBeAg-negative chronic hepatitis (moderate/high serum HBV DNA levels and elevated ALT) characterized by a higher risk of disease progression, the development of cirrhosis, and HCC [15]. A very recent study, analyzing, by whole-exome sequencing, HBV integration in liver tissue from 84 HBeAg-negative patients with low, moderate, and high serum HBV DNA, has shown that HBV integration occurs across all HBeAg-negative patients with CHB, including little-viremic patients with limited hepatic HBV reservoirs, and that by AUROC (Area Under the Receiver Operating Characteristics), HBsAg > 5 × 10^3^ IU/mL identified the occurrence of HBV integration in the whole exome with the best diagnostic accuracy (86.5%) [65]. The preservation of HBsAg production independently of HBV replication leads us to hypothesize its role in viral persistence. This assumption is supported by evidence demonstrating that high levels of HBsAg production can lead to T-lymphocyte exhaustion with consequent dysfunctional or weak T-lymphocyte responses. Moreover, secreted antigens may cause the clonal deletion of T lymphocytes [14]. Altogether, both functional and deletional tolerance may synergize in preventing the clearance of HBV infection. Thus, HBV integration may contribute to both sustaining the suppression of the HBV-specific immune response and supporting viral infection persistence. Indeed, as it has been shown that an inoculum of a single infectious virion is able to induce massive spread of the virus with the infection of virtually 100% of the hepatocytes [113], just a few HBV-infected cells are able to escape immunosurveillance and to maintain chronic infection in HBeAg-negative patients [8,114]. Integrated HBV DNA, as a source of HBsAg, may also provide an explanation for the lack of a positive correlation between HBsAg titers and serum HBV DNA and liver cccDNA levels in HBeAg-negative patients and in those under NA therapy [115,116,117,118,119,120,121,122]. This is in line with a study showing minimal change in HBsAg expression despite the achievement of undetectable cccDNA in patients treated with NAs for more than 5 years [116]. Moreover, HBsAg from integrated HBV DNA may help in explaining the differential response to first-generation RNA interference (RNAi)-based therapy targeting HBV transcripts, which was characterized by a lower magnitude of HBsAg reduction in HBeAg-negative or NA-experienced HBeAg-positive CHB patients in comparison to NA-naïve HBeAg-positive patients [37]. Preliminary results from studies testing a second generation of siRNAs targeting different regions of the HBV genome- indicate that the HBsAg reduction was similar in HBeAg-positive and negative patients [123]. Altogether, these results suggest that the complete immune control of HBsAg (produced from either cccDNA or integrated HBV DNA) is absolutely necessary to prevent hepatocyte reinfection and to achieve a functional cure.

Of additional potential clinical importance is the expression of HBx from integrated HBV DNA. The enhancer I/X promoter sequence is frequently present and active in HBx integrated sequences [124]. Moreover, upon integration, the 3’ end of the HBx protein is often deleted. Indeed, the 3′ end of the X gene corresponds to the region of the HBV genome where recombination breakpoints have the highest probabilities of occurring (located between the direct repeats DR1 and DR2). However, there is evidence showing that HBx mutants truncated up to 14 aa are still functional in transcriptional transactivation [36,125,126]. The functionally active HBx produced from integrated HBV may have a role in the establishment and persistence of viral infection, as HBx is critical for the HBV life cycle [34,127]. The efficient replication of HBV requires the regulatory HBx protein, which may promote viral replication both directly by acting on viral genome promoters and indirectly by an epigenetic modulation of the cccDNA minichromosome, and by affecting cellular pathways that favor virus replication [30,34,36,128,129,130]. In this regard, it is worth mentioning that many of the HBx activities supporting HBV replication may concurrently contribute to the HBV-associated carcinogenetic process [34,36]. It has also been highlighted that mutant HBV proteins expressed stably from integrated HBV DNA can be endowed with peculiar properties, which may exert ‘‘transacting’’ oncogenic function [34,36].

Many data indicate that mutated HBsAg can also be produced from integrated HBV DNA [126,131]. There is evidence suggesting that integrated HBV sequences may direct the nonphysiological expression of surface proteins [132,133,134]. It has been shown that hepatocytes harboring mutated large and/or middle surface proteins have a potential growth advantage. Hepatocytes overexpressing these proteins show the typical aspect of ground glass hepatocytes, and frequently cluster in nodules because of their higher proliferative activity and consequential clonal expansion [132,134,135]. The overproduction and accumulation of mutated large and middle surface proteins in the endoplasmic reticulum (ER) is a cause of significant ER stress and of the unfolded protein response, with the consequent induction of oxidative DNA damage [131,132,133,134,136] that can be converted to double-stranded breaks during hepatocyte regeneration [40]. Interestingly, it has been shown that mutated large surface protein may inhibit DNA double-stranded break repair and lead to genomic instability (Figure 2) [137]. As reported above, double-stranded breaks are the primary source of integrations in infected hepatocytes [40]. Thus, oxidative DNA damage induced by the intracellular accumulation of surface proteins may lead to increased amounts of HBV DNA integrated into the hepatocyte genome. This means that HBV itself may exert a relevant role in inducing host genome double-stranded breaks and HBV DNA integration (Figure 2).

### HBV Integration in Occult B Infection

The achievement of HBsAg loss and of a functional cure does not completely erase the risks associated with HBV cccDNA persistence and can lead to occult hepatitis B, which represents the fifth HBsAg-negative phase of chronic HBV infection [15,20,27,97]. HBV integration has also been detected in this phase of chronic infection [12,20,97]. However, most of the available data have been obtained from OBI patients who have developed HCC [16,20,55,102,138,139]. Very few studies have investigated HBV integration in OBI patients with chronic hepatitis related to different etiologic factors. Interestingly, studies conducted in patients with chronic hepatitis C infection and OBI have shown that the presence of integrated HBV DNA in these patients was associated with accelerated hepatocarcinogenesis [140,141]. Results obtained from nontumor liver tissues of OBI patients with HCC have revealed the presence in these tissue samples of HBx RNA and protein but not of S and Core transcripts and respective proteins [142,143,144]. Moreover, as mentioned above, it has been shown that HBx transcripts can be produced from both free HBV genome and integrated viral DNA, and that they are frequently truncated or highly mutated in tumors but not in surrounding nontumorous tissues [36,145]. Functional studies have highlighted the different properties of tumor-derived HBx proteins compared to wild-type HBx [36]. In particular, it has been shown that mutated HBx from HBsAg-negative HCC may retain a transactivating capability and the capacity to bind p53 and block p53-mediated apoptosis, thus conferring a growth advantage to neoplastic cells [146]. In addition, both in vitro and in vivo studies demonstrated that the C-terminal-truncated HBx, in contrast to the wild-type protein, could effectively induce the transformation of immortalized liver cells and affect the expression of critical genes involved in cell cycle control and apoptosis [147]. There is evidence demonstrating that the C-terminus of HBx affects the stability of the protein, its transactivation activity, and its ability to stimulate HBV replication [148], thus possibly favoring the silencing of cccDNA in OBI [145]. Furthermore, HBx mutants from HBsAg-negative HCC, harboring specific amino acid mutations or deletions of the C-terminal region, may be less effective in transcriptional transactivation, may induce both cell cycle arrest and the inhibition of apoptosis, and may also promote cell transformation [149].

HBV integration in patients with HBsAg-negative HCC have been reported since the early 1980s [47]. Subsequently, a number of studies performed with PCR-based assays or high-throughput sequencing approaches have confirmed this observation and have shown the presence of HBV insertion sites in 60–75% of HCC from OBI patients [55,102]. Interestingly, several studies have reported that HBV integration frequently occurs in OBI patients who have developed HCC in the absence of cirrhosis [54,56,102,140], also supporting a cis-acting role of HBV integration in the tumorigenesis process for patients with OBI. Indeed, sarco-/endoplasmic reticulum Ca^2+^ ATPase (which pumps calcium from the cytosol to the endoplasmic reticulum) and *PARD6G* (the partitioning-defective-6-homolog-gamma gene encoding a protein that is part of the Par6 complex, which is involved in the establishment of cell polarization and in the polarized migration of cells) have been reported in HCC in noncirrhotic HBsAg-negative patients following HBV DNA integration [54,56].

## 5. Conclusions

During the last few years, knowledge in the HBV DNA integration field has greatly improved owing to the development of new detection methods and research models. However, some key aspects of viral integration remain to be defined, such as the role of HBV integration in the pathogenesis of liver disease, in HCC initiation and development, as well as its impact in the achievement of a functional cure. HBV DNA integration appears to occur early in viral infection. It has been detected at the acute stage of HBV infection, in subjects with a severe form of acute hepatitis, and in the early “immune-tolerant” phase of chronic infection. During chronic hepatitis, particularly in the HBeAg-negative phase, hepatocytes with integrated HBV DNA undergo extensive selective clonal expansion. Interestingly, the frequency of integrated DNA in the liver appears to remain stable during virus clearance by immune or antiviral therapy. Thus, the cell genomic alterations acquired as a consequence of chronic or acute viral hepatitis accumulate during and persist after the resolution of HBV infection. HBV-specific immune responses in the liver microenvironment as well as hepatocyte accumulation of HBsAg may favor HBV DNA integration via oxidative stress, increased levels of double-stranded DNA breaks, and clonal hepatocyte expansion. Although no significant enrichment of any specific structural or functional host genomic element has been observed during such clonal hepatocyte expansion, the evidence that viral DNA integrations in CHB can be detected at a frequency of one or more copies per hepatocyte implies that liver cells might also harbor a very high number of viral integrations [40], thus assigning a major role to DNA damage in the pathogenesis of hepatitis B liver disease and in the initiation of HCC.

On the basis of the available evidence, it appears that during chronic liver infection, enhanced hepatocyte turnover due to necroinflammation and immune selection against HBV-replicating hepatocytes leads to the loss of some hepatocyte lineages and the expansion of others over time. Clonally expanded hepatocytes with selective proliferative advantages containing integrated HBV DNA may be prone to acquiring additional genetic mutations or epimutations that are positively selected for in the altered microenvironment of an injured liver. Consequently, the “mutant clone” can expand to produce fields of cells that are predisposed to eventually progress to cancer (Figure 2). Field cancerization is recognized to underlie the development of many types of cancer, including lung, colon, skin, prostate, and bladder cancer [150]. Progression towards HCC more likely involves the accumulation of multiple field cancerization mutations, mostly at the cirrhotic stage of liver disease. Epistasis among a synergistically acting group of mutations might be the cause of phenotypic change. Indeed, each mutation, including those induced by HBV integration, would not be sufficient to cause a change to a cancerous phenotype without the others. Phenotypic changes (which do not necessarily entail morphological changes) could imply the acquisition of properties such as an increased growth and survival rate, decreased death rate, and increased immune escape and may culminate in the formation of dysplastic nodules, which are recognized as preneoplastic lesions [151]. Mutations in the *TERT* promoter are among the initial signs of neoplastic transformation, as they can already be detected in dysplastic nodules [152]. The *TERT* promoter is a recurrent insertion site for HBV DNA [53,61]. To date, the activation of *TERT* along with the alteration of the *CCNA2*, *CCNE1*, *MLL4*, *SERCA1*, and *PARD6G* genes by HBV insertions is among the few recognized early events through which HBV may trigger hepatocarcinogenesis, since these phenomena may occur in the absence of cirrhosis [53,55,56,61,66,102,141] and other oncogenic events such as *p53* or *CTNNB1* mutations [153].

As an integral element of the host genome, integrated HBV DNA is more stable than cccDNA. Many direct-acting antivirals and host-targeting agents that have, as an ultimate goal, the elimination or silencing of cccDNA in order to achieve a functional cure of HBV infection have been developed. However, none of these therapeutic agents address the issue of integrated HBV DNA. In the setting of a functional cure, integrated HBV DNA persists and can continue to produce HBsAg. Several obstacles still hinder the achievement of a functional cure. These include cccDNA, which is not affected by existing therapies; T-cell immune exhaustion, which may be mediated by continuous HBsAg production; and integrated HBV DNA, which can act as a stable template for the expression of viral proteins. However, whether integrated HBV represents a barrier to achieving a functional cure remains a matter of debate. Further studies clarifying how integrated HBV sequences can affect host genome stability, promoting the progression of liver disease, and how their transcription products can damage liver cells are essential to gain a better understanding of HBV pathobiology and for the development of curative therapies.

## Figures and Tables

**Figure 1 viruses-13-00368-f001:**
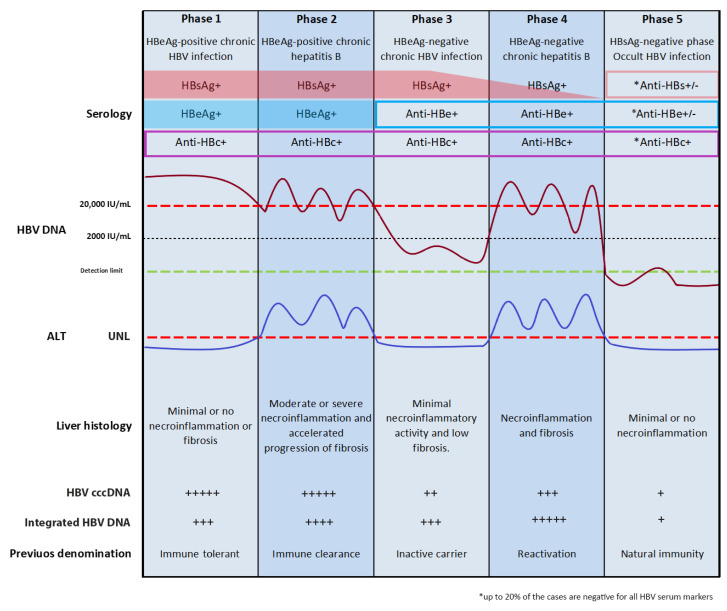
Natural history of hepatitis B. The natural history of chronic hepatitis B has been schematically divided into five clinical or virological phases (according to European Association for the Study of the Liver 2017 guidelines); +, rarely detected; ++, occasionally detected; +++, often detected; ++++, frequently detected; +++++, almost always detected.

**Figure 2 viruses-13-00368-f002:**
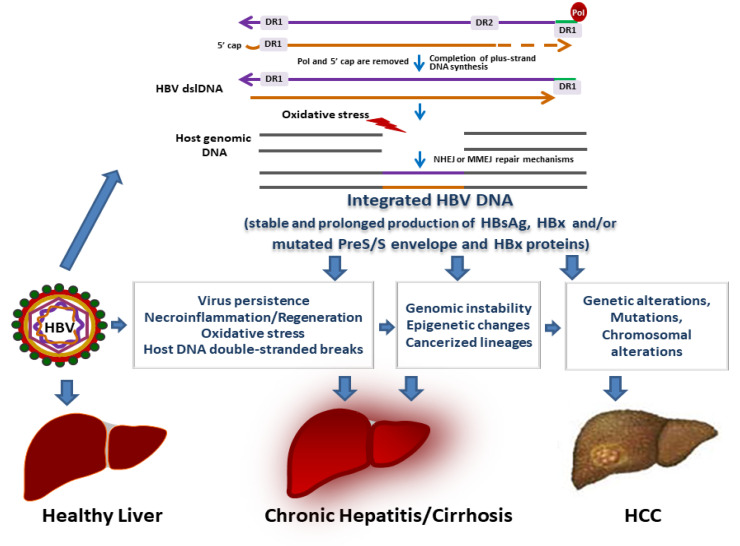
Role of HBV DNA integration in liver disease and HCC development. The HBV double-stranded linear DNA (dslDNA) obtained after the plus-strand DNA extension is completed is the main substrate for HBV DNA integration. HBV dslDNA can integrate into the host cell genome at the sites of cellular double-stranded DNA breaks by nonhomologous end joining (NHEJ) or microhomology-mediated end joining (MMEJ) repair pathways depending on the sequence characteristics at the ends of the viral DNA and the host double-stranded break DNA. Integration of HBV DNA sequences into the host genome as well as prolonged production of HBx and HBsAg and/or of modified forms of HBx and preS/S envelope proteins may contribute to viral infection persistence and to liver damage via multiple mechanisms, which ultimately lead to liver cancer development.

**Table 1 viruses-13-00368-t001:** Detection methods for HBV DNA integration.

Technique	Suitable Uses	Advantages	Limitations	References
Southern blot hybridization	HBV DNA integration detection in liver samples with highly expanded hepatocyte clones	Low cost	➢ Time consuming ➢ No sequence information ➢ Dependent on restriction enzyme sites ➢ Low sensitivity	[47,48,49,50]
Direct cloning and Sanger sequencing	Defining structure of HBV integration in liver samples with highly expanded hepatocyte clones	Definition of the sequences of integrated HBV DNA and the adjacent cellular DNA	➢ Not suitable for screening a large number of unknown virus–cell junctions ➢ Low throughput ➢ Dependent on restriction enzyme sites for cloning	[50,51]
Alu PCR	Detecting and sequencing of HBV integration in clonally expanded hepatocytes	Inexpensive	➢ Can only effectively detect HBV integration near Alu sequences ➢ Dependent on Alu sequences ➢ No integration quantification ➢ Biased towards larger clones	[52,53,54,55,56,57]
Inverse PCR	Detecting and quantifying HBV DNA integrations in small hepatocyte clones (single copy virus–cell junctions can be detected)	➢ Absolute integration quantification ➢ High sensitivity	➢ Dependent on restriction enzyme sites for detection of virus–host DNA junction ➢ Detection of integrations that occur between nucleotides ~1650 and ~1850 of viral genome	[39,41,58,59,60]
Whole genome sequencing (WGS)	Sensitive and comprehensive inthe identification of viral integrants across the human genome	Full genome coverage	➢ Low depth ➢ High cost ➢ No absolute quantification	[61,62,63]
Whole-exome sequencing	Detection of HBV integration in coding regions	Greater depth than WGS	➢ Coverage limited to coding regions ➢ No absolute integration quantification	[64,65,66,67]
RNA Sequencing	Sensitive and comprehensive in the identification of viral integrants across the human transcriptome	➢ Greater depth than WGS ➢ Data on transcriptional activity	➢ Coverage limited to expressed coding regions ➢ No absolute integration quantification ➢ Biased towards more highly expressed genes	[42,65,66,67,68]
Capture-enriched next generation sequencing	High-throughput viral integration detection method	➢ Cost-effective compared with WGS ➢ Quicker and less laborious than PCR-based methods	➢ Lower sensitivity than PCR-based methods ➢ Lower specificity than PCR-based methods ➢ Shorter fragments are captured with higher specificity than longer ones	[42,69]

## Data Availability

Data sharing not applicable.

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
