# Peer review of "HBV-Integration Studies in the Clinic: Role in the Natural History of Infection"

_viruses, 2021, doi:10.3390/v13030368_

Round 1
Reviewer 1 Report
Comprehensive review of HBV integration and how it relates to clinical states. Minor suggestions: p 5, line 213: It should be clear that these 5 phases are per EASL guidelines since the phases are not referred to in this way universally yet. You state later on that these are based on EASL but it should state that here as well.
In Figure 1, tolerant is spelled incorrectly.
Reviewer 2 Report
This is an extremely well written and clear review aimed at providing a comprehensive overview of the role of HBV-DNA integrations during the natural history of chronic HBV infection and its potential implications in the pathogenesis of infection. There are some minor points to be addressed:
1.The authors mentioned that HBV-DNA integrations can be detected by different methodologies. It would be interesting to include a Table reporting such methodologies with related advantages and disadvantages.
2.The authors reported that HBV-DNA integrants can localize in transposable elements. At this regard, it would be nice to discuss two manuscripts: the former by Ramesh Rajaby et al., Nucleic Acid Res 2021 highlighting that HBV integrations are enriched in the previously overlooked LINE and Satellite regions and the latter by Liang et al., J Hepatol 2016 on the role of the chimeric transcript HBx-LINE1 in HBV-driven hepatocarcinogenesis.
- Please, discuss, in the paragraph related to HBV-DNA integrations in HBeAg-negative phase, the recently published study by Svicher et al., Gut 2021 focusing on the characterization of HBV-DNA integrations profiles across HBeAg-negative patients with different levels of serum HBV-DNA.
- Please, discuss the work, presented at the digital international liver conference (EASL 2020), by Ramirez R, et al. J Hepatol 2020;73:S6–7, highlighting that HBV integrations are not restricted to partial regions, but also involve the full viral genome, supporting the contribution of integrated HBV DNA to human genomic alterations.
